# Machine learning-based motion tracking reveals an inverse correlation between adhesivity and surface motility of the leptospirosis spirochete

Keigo Abe[1], Nobuo Koizumi [2] & Shuichi Nakamura [1]✉

Bacterial motility is often a crucial virulence factor for pathogenic species. A common approach to study bacterial motility is fluorescent labeling, which allows detection of individual bacterial cells in a population or in host tissues. However, the use of fluorescent labeling can be hampered by protein expression stability and/or interference with bacterial physiology. Here, we apply machine learning to microscopic image analysis for label-free motion tracking of the zoonotic bacterium *Leptospira interrogans* on cultured animal cells. We use various leptospiral strains isolated from a human patient or animals, as well as mutant strains. Strains associated with severe disease, and mutant strains lacking outer membrane proteins (OMPs), tend to display fast mobility and reduced adherence on cultured kidney cells. Our method does not require fluorescent labeling or genetic manipulation, and thus could be applied to study motility of many other bacterial species.

The locomotive ability of bacteria allows them to actively explore environments favorable for their growth, facilitating their proliferation. Though diverse forms of bacterial motility are known, the flagellum has been the primary machinery. The rotation of extracellular flagella possessed by, for example, *Escherichia coli*, *Salmonella enterica*, and *Vibrio cholera*, propels the bacteria through interaction with fluids[1–3]. Some species employ extracellular flagella for swarming over surfaces covered with viscous agents[4]. When the species are causative agents of diseases, motility is usually involved in invasion, migration within the host, and disease development[1,2,5]. Thus, understanding the responsibility of motility as a virulence factor will lead to the development of medications and the proposal of novel strategies for protecting against infection.

Leptospirosis is an emerging zoonosis spreading globally. Regarding the threat to human health, the spirochete pathogens of leptospirosis, *Leptospira* spp., have caused ~1 million severe cases and ~60,000 deaths yearly[6]. Pathogenic *Leptospira* spp. have been classified into more than 250 serovars, and clinical signs in infected animals differ depending on the combination of host species and *Leptospira* serovars[7–9]. Susceptible animals, including humans, can develop severe manifestations, such as jaundice, pulmonary hemorrhage, and nephritis, whereas rodents are asymptomatic carriers for particular serovars. Some animals, such as dogs, maintain the host-adopted pathogens after recovery from the disease. *Leptospira* spp. colonize in the kidneys of particular reservoir animals and are excreted in the urine. Bacteria excreted into the environment infect animals through the percutaneous or permucosal route. In the host body, *Leptospira* spp. migrate to specific organs and break tissues[7–10]. *Leptospira* spp. have two flagella in the periplasmic space[11]. It is believed that the rotation of the periplasmic flagella (PFs) beneath the outer membrane drives the rolling of the spiral body[12]. The PF-dependent rotation of the cell body produces the thrust needed to swim in liquid media and crawl on a solid surface (Fig. 1). In other words, spirochetes have an amphibious motility system[13]. Animal experiments have shown that motility-deficient mutants were attenuated[14,15], suggesting that leptospiral virulence likely involves

[1]Department of Applied Physics, Graduate School of Engineering, Tohoku University, Sendai, Miyagi, Japan. [2]Department of Bacteriology I, National Institute of Infectious Diseases, Tokyo, Japan. ✉e-mail: shuichi.nakamura.e8@tohoku.ac.jp

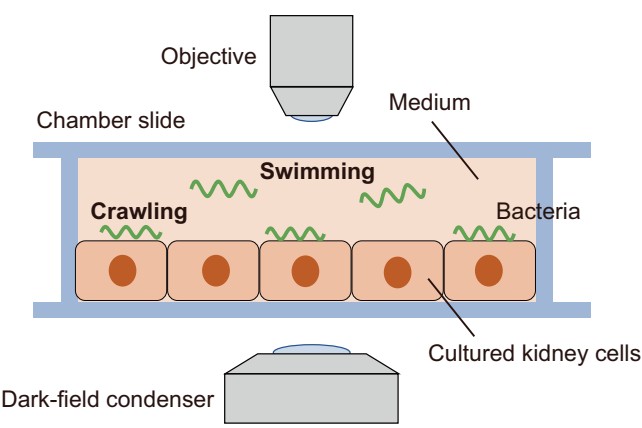

**Fig. 1 | Observation of bacterial locomotion over cultured cells.** Cultured kidney cells form a monolayer on the glass of a chamber slide filled with a liquid medium. Leptospirosis spirochetes swim in the medium and crawl on the kidney cells.

motility. However, the beneficial role of motility for leptospiral pathogenicity remains unclear.

Direct observation of bacteria in various conditions, such as in solutions, tissues, and artificial microstructures, has been the conventional method for investigating disease mechanisms in terms of pathogen dynamics and motility mechanisms[16–19]. These experiments usually pursue representative bacteria in a crowd or detect bacteria dwelling in living cells, which requires labeling the objects with fluorescent proteins or dyes. However, not all bacterial strains can endure genetic engineering and express fluorescent proteins. Labeling can possibly affect bacterial physiology and dynamics. We previously used a green fluorescent protein (GFP) to observe *Leptospira* spp. over the cultured kidney cells. The experiment showed that the pathogenic species *L. interrogans*, which causes leptospirosis, moved more actively than the saprophytic *L. biflexa*[16]. We attempted to label several pathogenic strains with a GFP, but only two strains yielded fluorescence and were available. Hence, label-free techniques allow experiments that are not restricted by the properties of biological samples.

In this study, we developed a method for label-free motion tracking to elucidate the mechanism by which leptospirosis spirochetes move on host cells. Our label-free bacterial tracking method uses machine learning-based image processing that highlights moving objects by subtracting the background from raw images. The processed images allowed label-free analysis of bacteria motility on the cultured animal cells, providing insight into the mechanism of leptospiral locomotion on host cells and the significance of motility as a virulence factor.

## Results

### Label-free motion tracking

We used a dark-field microscope to observe a thin cell body of *L. interrogans* (~100 nm in width). In the in vitro infection assay (Fig. 1), scattering from cultured cells prevents the visualization of bacteria (Fig. 2a, Step 1 and Supplementary Movie S1). Therefore, we first obtained brightness distributions of individual pixels in the time axis direction and then subtracted the background cluster from the raw image by the pixel (Fig. 2a, Step 2, 2b, and Supplementary Movie S2). We recognized the background (mainly signals from cultured cells) in the complex distribution using a Gaussian mixture model (GMM). The GMM was sequentially updated every time a new pixel value was added through maximum a posteriori (MAP) estimation (Fig. 2c, d)[20,21]. The background subtraction results processed by the pixel were integrated (Fig. 2a, Step 3 and Supplementary Movie S3), and bacteria were recognized according to area and shape (Fig. 2a, Step 4 and Supplementary Movie S4). As such, we successfully determined the

locomotion trajectories of bacteria without labeling by tracking their centroids based on the frame (green lines in Fig. 2a, Step 5 and Supplementary Movie S5).

### Confined crawling in association with reservoir host cells

We first compared the *Leptospira* swimming and crawling motility between two pairs of the leptospiral strain and host species. One of the pairs, *L. interrogans* serovar Manilae vs. NRK (normal rat kidney) cells, was used as an example of asymptomatic because rats are known to be typically asymptomatic carriers of serovar Manilae strains[22]. The other pair, *L. interrogans* serogroup Hebdomadis strain D-OW16-2K (isolated from a seriously ill dog) vs. MDCK (Madin−Darby canine kidney) cells, was assumed to develop severe disease. In the asymptomatic pair, leptospires swam smoothly, but their translational locomotion was reduced while crawling in association with the kidney cells (Fig. 3a, left panels). In contrast, leptospires in the severely symptomatic pair showed long-distance migration on the cell surface (Fig. 3a, right panels). We previously observed crawling of GFP-labeled *L. interrogans* on kidney cells, showing higher mobility of pathogenic strains than a non-pathogenic strain[16]. The present label-free experiment showed a similar tendency.

The previous study using GFP-labeled strains showed that the crawling speed of leptospires was not associated with disease severity[16]. However, the label-free experiment revealed that leptospires swam faster in the asymptomatic pair than in the severely symptomatic pair (Fig. 3b, upper panel) but that the opposite was true for crawling speed (Fig. 3b, lower panel). We also evaluated the two-dimensional movements of leptospires according to the time plot of the mean square displacement (MSD) $<r^2(\Delta t)> = <|\mathbf{r}(t+\Delta t)-\mathbf{r}(t)|^2>$ (Fig. 3c). It is known that the $<r^2(\Delta t)>$ of particles moving unidirectionally is proportional to $\Delta t^2$, whereas that of diffusive motion is proportional to $\Delta t$[23]. Every $<r^2(\Delta t)>$ plot of leptospires showed a quadratic curve at a very short time scale ($\propto \Delta t^2$) and a proportional line at a longer time scale ($\propto \Delta t$. See also log−log plots). The observed time dependences of $<r^2(\Delta t)>$ represent bacteria moving in a direction for a short duration and those showing diffusion for an extended duration. A swimming analysis of *E. coli* proposed a model determining the duration for smooth unidirectional locomotion: $<r^2(\Delta t)> = 2v_0^2\tau[t-\tau(1-\exp(-\Delta t/\tau))]$, where $v_0$ is the speed at which the bacterium swims in one direction and $\tau$ is the duration of the ballistic swimming[24,25]. Applying the model to the leptospiral locomotion showed that $\tau \simeq 1$ s. The predicted duration for ballistic swimming agrees with the change in the time dependence of $<r^2(\Delta t)>$ observed in Fig. 3c. We determined the diffusion coefficients of swimming ($D_s$) and crawling ($D_c$) using linear regression to the regime proportional to $\Delta t$. Accordingly, we found that leptospires in the asymptomatic pair diffused more readily by swimming than those in the severely symptomatic pair but that the opposite was true for the magnitude of diffusion by crawling (Fig. 3c). These results suggest that the confined behavior of leptospires on tissue cells enables host animals to carry the bacteria without apparent clinical signs.

### Leptospires crawl faster on dog kidney cells

Focusing on speeds, we examined the motility of various *Leptospira* strains (Supplementary Table S1) over NRK and MDCK. Swimming speed does not seem to be related to the host-bacteria combination, but crawling speed is likely to be faster for strains infected to MDCK (Fig. 4a).

Such a tendency could be seen more clearly by comparing the speeds measured on MDCK and NRK (Fig. 4b). Most of the measured strains showed faster crawling on MDCK (MDCK/NRK > 1). Dogs are susceptible to many pathogenic *Leptospira* serovar strains, and in some cases serious illness develops[8]. Although virulence of serovar Manilae strain in dogs remains unclear, anti-serovar Manilae antibodies were frequently detected in dogs in the Philippines, indicating the

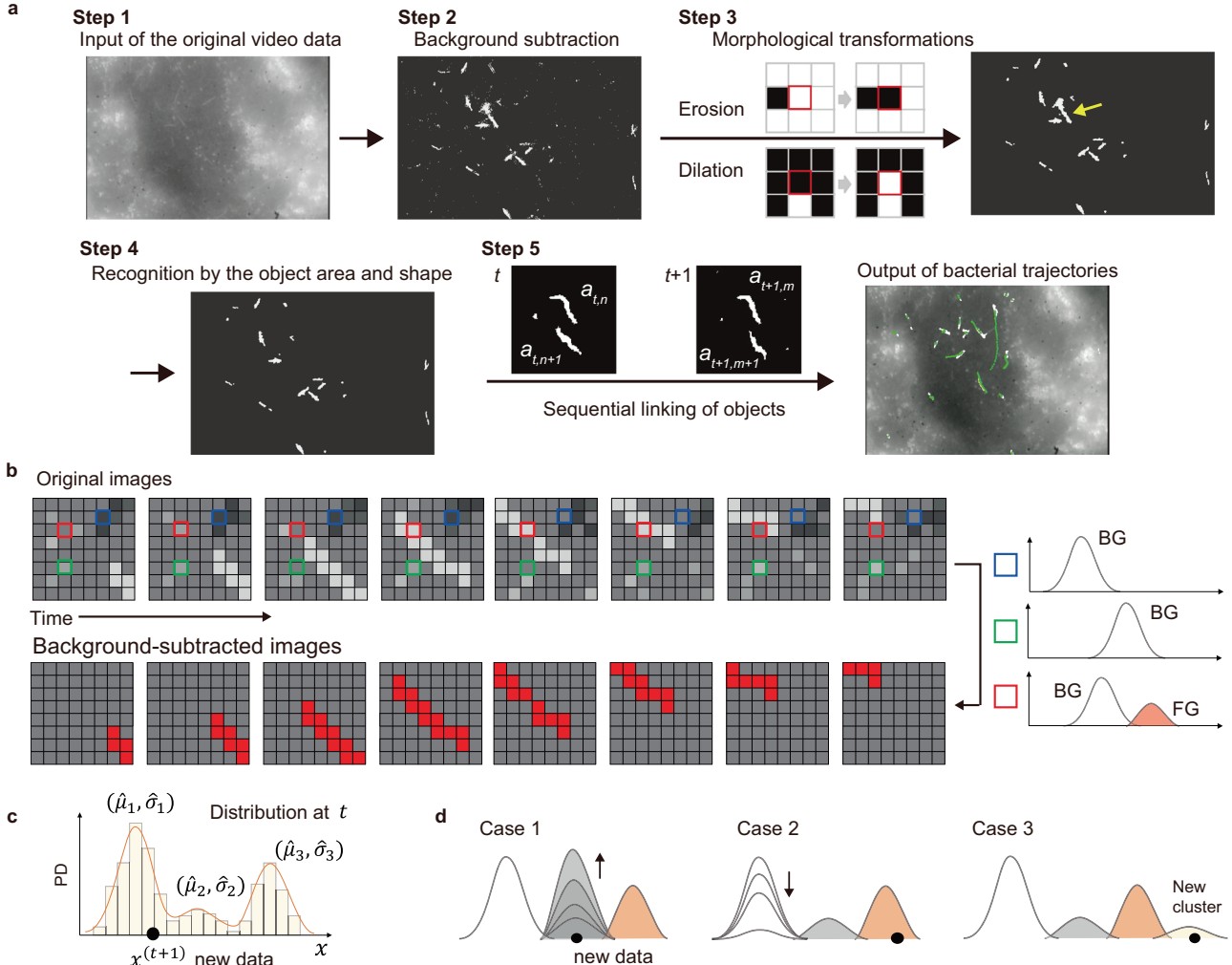

**Fig. 2 | Label-free bacterial tracking on cultured cells using machine learning.** **a** An overview flowchart. A mass of overlapped bacteria observed in Step 3, indicated by a yellow arrow, is removed by Step 4. **b** A schematic explanation of background subtraction. Sequential measurements of the pixel brightness along the time axis yield brightness distributions by the pixel. Focus on three pixels indicated by blue, green, and red squares. The blue and green pixels show single clusters with different mean brightness. The red pixel shows two clusters caused by the invasion of a brighter object. Major and minor clusters generally correspond to background (BG) and foreground (FG) signals, respectively, which are thus discriminated without setting the uniquely specified threshold. **c** Update of GMM upon the addition of a new data. The new data $x^{(t+1)}$ is classified based on the Mahalanobis distance $D_m$ from the clusters comprising the distribution at $t$: $D_m(x^{(t+1)}) = |x^{(t+1)} - \hat{\mu}_m|/\hat{\sigma}_m$, where $\hat{\mu}_m$ and $\hat{\sigma}_m$ are the mean and variance of the component $m$. **d** Anticipated GMM updates. Adding new data could cause an increase (Case 1) or decrease (Case 2) in existing clusters or the generation of a new one (Case 3). If $D_m(x^{(t+1)})$ is less than the threshold, then $x^{(t+1)}$ is classified to the $m$th cluster; if $D_m(x^{(t+1)})$ is larger than the threshold (i.e., there are no close clusters), then a new $(m+1)$th component is generated with $\hat{\pi}_{m+1} = \alpha$, $\hat{\mu}_{m+1} = x^{(t+1)}$ and $\hat{\sigma}_{m+1} = \sigma_0$, where $\alpha$ decays the past data, $\hat{\pi}_m$ is the weighting parameter (non-negative, add up to 1), and $\sigma_0$ is some appropriate initial variance. The minor cluster vanishes when the number of clusters exceeds the upper limit set arbitrarily.

infectivity of the serovar to dogs[26]. Among the strains measured, some strains, such as serogroups Australis and Autumnalis (Autum), showed no significant enhancement of crawling. Therefore, these results suggest that crawling is not a sole determinant of virulence but is involved in the pathogenicity of *Leptospira* along with many other virulence factors.

## OMPs contribute to crawling

The crawling of the periodontal spirochete *Treponema denticola* involves dentilisin, a protein located on the outer membrane[27]. Studies on *Leptospira* crawling also showed the possible contribution of outer membrane proteins (OMPs)[28] and lipopolysaccharides (LPS)[13] to surface locomotion, but the responsible *Leptospira* cell surface molecules remain unidentified. Note that the leptospiral OMPs LenA and LigA have the ability to bind to the host cells[29–32]. Examining the effect of the deficiency of LenA and LigA on the adhesion of *L. interrogans* serovar

Manilae, we found that both disruptions reduced the leptospiral adherence to NRK cells (Fig. 5a).

The malfunction of LenA and LigA affected the swimming speed (Supplementary Note 1 and Fig. S1). The swimming speed is given by free cell rotation without interaction with surfaces (equivalent to $v_0$ in the crawling model described later). We thus evaluated the crawling ability by normalizing the crawling speed $v_{CR}$ with the swimming speed $v_{SW}$: $v_{CR}/v_{SW}$. The crawling ability of *L. interrogans* serovar Manilae on its reservoir's cells (NRK) was facilitated by the LenA and LigA deficiency (Fig. 5). These results suggested that LenA and LigA could be involved in adhesion and crawling along with LPS and other unknown adhesins. LenA binds to the complement regulatory protein factor H, suggesting the contribution of the protein to immune evasion[29]. LigA has the ability to bind to fibrinogen and extracellular matrix proteins (e.g., collagen and laminin)[31]. The targeted repression of the *ligA* expression attenuated *L. interrogans* serovar Manilae, suggesting the

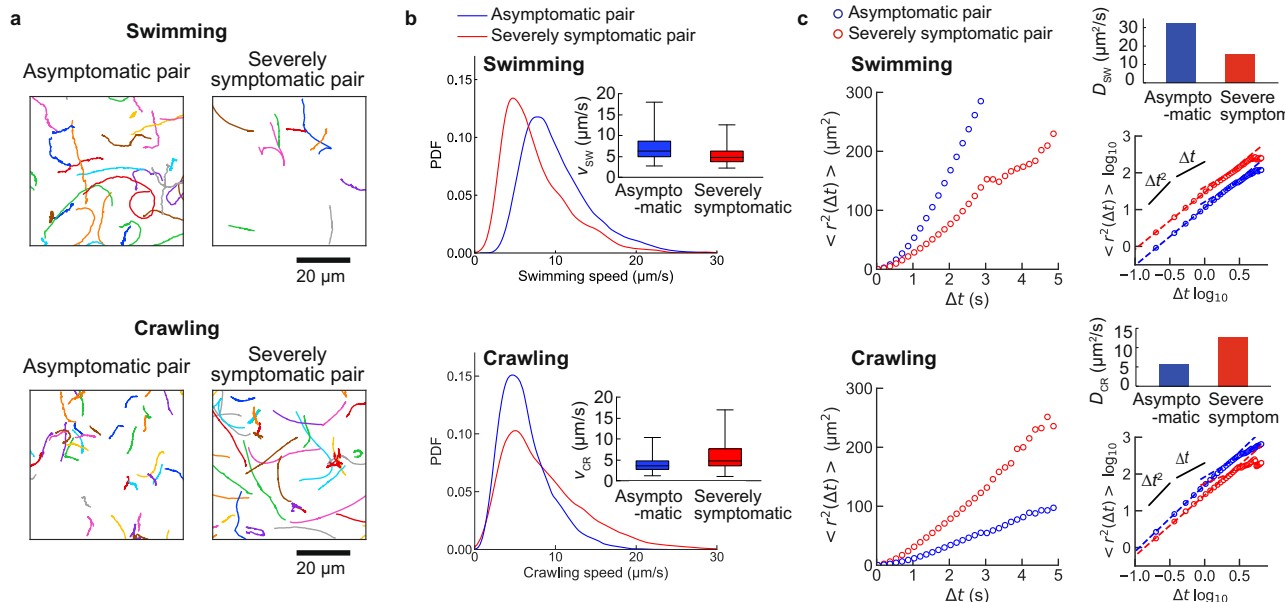

**Fig. 3 | The motion of the leptospirosis spirochetes over kidney cell monolayer.** Swimming bacteria were measured in liquid phases over kidney cells (Fig. 1). **a** Swimming and crawling trajectories of representative bacteria are presented in 3-s runs. The analyzed trajectories for swimming and crawling were 182 and 133 in the asymptomatic pair and 20 and 127 in the severely symptomatic pair. **b** Probability distribution functions (PDFs) of swimming and crawling speeds. The box-and-whisker plots (insets) show the 25th (the bottom line of the box), 50th (middle), and 75th (top) percentiles and the minimum and maximum values (whiskers) of the speed data set obtained in each pair. The measured bacteria were 123 (swimming of asymptomatic), 23 (swimming of severely symptomatic), 125 (crawling of asymptomatic), and 84 (crawling of severely symptomatic). **c** Time plot of the mean square displacement (MSD). The log–log plots show the ballistic ($\propto \Delta t^2$) and diffusive ($\propto \Delta t$) regimes; the time dependence changes at ~1 s (see also the main text). The upper right panels show the diffusion coefficients determined by the line fitting to the regime $\propto \Delta t$.

significance of LigA as a virulence factor[33]. On MDCK cells, the crawling ability of *lenA* mutant was also enhanced, whereas few adherent and crawling cells were observed in the *ligA* mutant (Supplementary Note 1 and Fig. S2), suggesting that LigA might be a crucial adhesin for the leptospiral colonization to dogs.

### Inverse correlation between adhesion and crawling
We examined the relationship between the *Leptospira* adhesivity and crawling ability $v_{CR}/v_{SW}$. Figure 6a shows an inverse correlation between adhesion and $v_{CR}/v_{SW}$. In agreement with Fig. 4b, leptospires tend to adhere more strongly to NRK cells, suppressing crawling.

When a bacterium is propelled at the velocity of $v$ by the rotation of helical flagella or helical cell body with the wavelength of $\lambda$ at the frequency of $f$, the conversion efficiency from the rotation to translation can be evaluated by $(v/\lambda)/f$. The value indicates the degree of slippery. For example, *Vibrio alginolyticus* shows $(v/\lambda)/f \simeq 0.07$, indicating that the bacterium migrates by only 7% of the flagellar wavelength upon one flagellar revolution[34]. We have shown that *L. biflexa* showed $(v/\lambda)/f \simeq 0.2$ when swimming and $\simeq 1$ when crawling ($\lambda \simeq 0.6\,\mu m$)[13]. Using the reported values of $(v/\lambda)/f$ and $\lambda$ together with the speeds measured in this study ($v_{SW}$ and $v_{CR}$), we predicted the cell body rotation rates during swimming ($f_{SW}$) and crawling ($f_{CR}$). In Fig. 6b, the host-bacteria pairs were arranged in order of adhesivity, showing an tendency of $f_{CR}/f_{SW}$ increasing with the decrease in adhesivity. This suggests that the sluggish crawling speed can be attributed to the deceleration of the cell rotation due to the strong adhesion to kidney cells. From the aspect of pathogenicity, strong adhesivity of bacteria would be advantageous to be maintained in the host, while weak adherence enables bacteria to explore a wide range of the host tissue. For *Leptospira* spp., fast and diffusive crawling would gain a chance to penetrate the deeper side of the tissue[10,18], while they have a risk of elimination by a flow of body fluid. These are likely to be a trade-off relationship that pathogenic bacteria are imposed, which is as well as physiological constraints observed between growth and survival[35], and migration and colonization[36].

### Crawling model
We demonstrate *Leptospira* crawling using a simple two-state model (Fig. 7a)[37,38]. Crawling occurs through successive binding and detachment of adhesins against the host cells with rate constants $k_{on}$ for binding and $k_{off}$ for detachment. A bacterium has $N_{ad}$ adhesins in total. Note that the effective $N_{ad}$, $N'_{ad}$, is defined by the number of host receptors per bacterial length $N_r$: If $N_{ad} \ll N_r$, $N'_{ad} = N_{ad}$; if $N_{ad} > N_r$, $N'_{ad} = N_r$. The bound adhesins are linked with the host cells via a spring with the spring constant $\kappa$, and $k_{off}$ depends on the extension of the spring $x$: $k_{off}(x) = k'_{off}\exp[\kappa x/\kappa l]$, where $l$ represents the typical length at which the bound OMP is released[38]. The force balance of motion in low Reynolds numbers, where the inertia can be negligible, is described by $F - \gamma v_{CR} - \kappa \sum_{i=1}^{N'_{ad}} x_i = 0$, where $F$ is the propulsive force produced by a *Leptospira* bacterium, $\gamma$ is the viscous drag coefficient, $v_{CR}$ is the crawling velocity, and $x_i$ is the spring extension of the $i$th adhesin. Therefore, $v_{CR} = (F/\gamma) - (\kappa/\gamma)\sum_{i=1}^{N'_{ad}} x_i$. Scaling $v_{CR}$ by the velocity without bonds $F/\gamma$ ($\equiv v_0$) yields $v_{CR}/v_0 = 1 - (\kappa/F)\sum_{i=1}^{N'_{ad}} x_i$. Here we assumed that $\kappa/F = 1$ for qualitative discussion.

Our experiments suggested that excessive adherence of leptospires decreased the crawling speed. Moreover, the combination of leptospiral strains and host species affected adherence and crawling (Fig. 6). A plausible cause for these phenomena could be a difference in kinetic parameters by the bacterium-host combination. The two-state model assuming $N'_{ad} = N_{ad}$ qualitatively demonstrated that an increment of $k'_{off}/k_{on}$ decreased the fraction of the bound adhesins $n_{on}/N'_{ad}$ (Fig. 7b), affecting $v_{CR}/v_0$ anti-correlatively (Fig. 7c). *L. interrogans* serovar Manilae might be adapted to rats in terms of stronger adhesin binding to the ligands on the kidney cells. We also found that the reduced number of OMPs by gene knockout facilitated crawling (Fig. 5). The tendency was shown by the model in the $N'_{ad}$ dependence

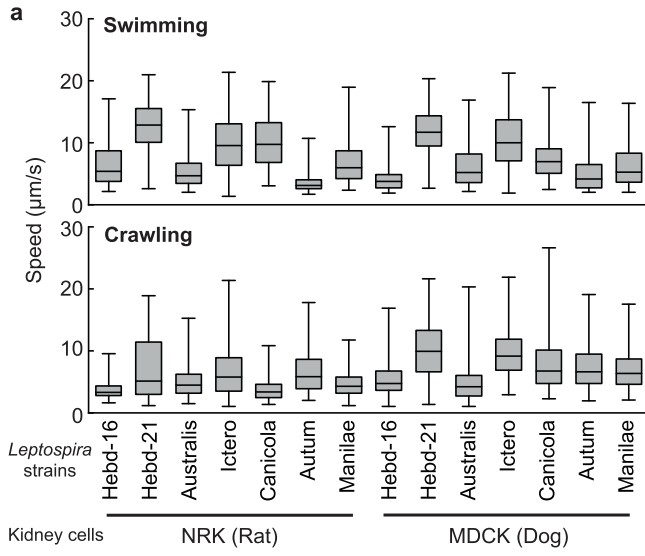

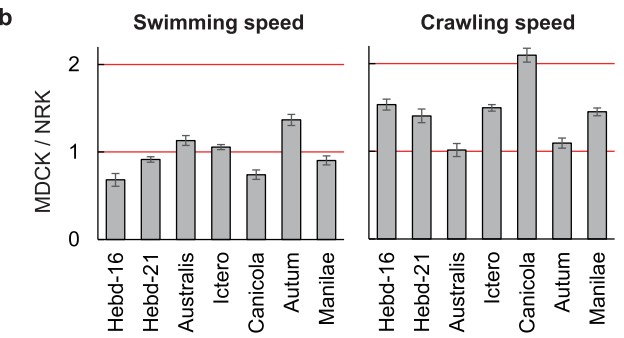

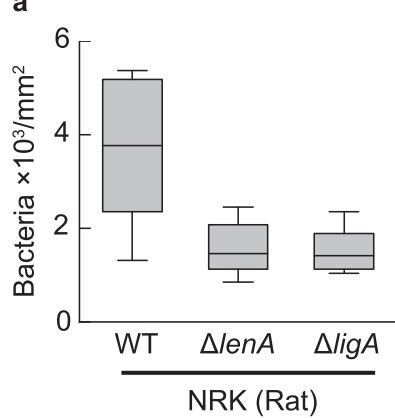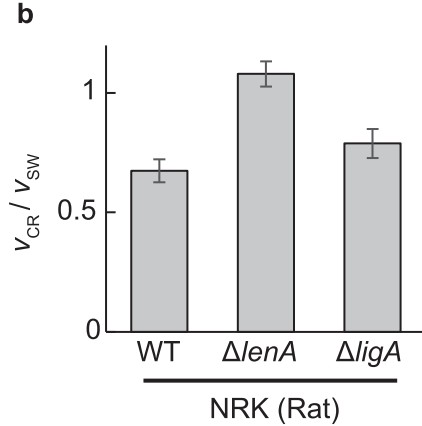

**Fig. 4 | Motility of various *Leptospira* strains on NRK or MDCK cells. a** Swimming and crawling speeds. Percentiles (boxes) and the minimum and maximum values (whiskers) of the speed data set are shown as described in the legend of Fig. 3. **b** The ratio of speeds measured using MDCK and NRK. The values of ratio and error were obtained from the averaged speeds and standard errors of the data set shown in a (see Supplementary Note 1 for details on error propagation). The strain names are presented by serovar names, and some of them are abbreviated: Hebd Hebdomadis, Ictero Icterohaemorrhagiae, Autum Autumnalis. Full names and the number of samples measured are shown in Supplementary Table S2.

**Fig. 5 | Effect of the loss of functional OMP genes on adhesion and crawling. a** Adhesion of the wild-type strain (WT) and mutants (Δ *lenA* and Δ *ligA*) of *L. interrogans* serovar Manilae to rat kidney cells. Percentiles (boxes) and the minimum and maximum values (whiskers) of the speed data set are shown as described in the legend of Fig. 3. **b** Crawling ability was evaluated by normalizing crawling

of $v_{CR}/v_0$ (Fig. 7c). Given that $N_r$ is likely not uniform in living cells and $N_{ad}$ can differ among bacteria, the bond strength acting on bacteria would accidentally change with locomotion (Fig. 7d, upper panel). This would be observed as speed fluctuation of crawling (Fig. 7d, lower panel). Spatial-temporal variation of $N'_{ad}$ may promote diversity in bacterial dynamics and ecology within the host body.

Quantitative modeling using practical values of parameters will be a subject for future studies. For example, the kinetic property of adhesins, assumed to be unity in the present model, is obviously diverse in the actual bacteria and host cell combinations[32]. $\kappa$ and $\gamma$ determine the relaxation time of the system by $\gamma/\kappa$. $\kappa$ could be characterized by the interaction between bacterial adhesins and receptors on the host cells, whereas $\gamma$ is governed by the environmental viscosity and bacterial morphology (especially size). Viscosity may differ according to bacterial habitat within the host body. Leptospires, which have a wide cell length range (~5 μm in the shortest and >20 μm in the longest), are observed in a population. Also, $F$ is related to torque produced by the flagellar motor[39,40]. Although some of the values will be obtained as a fitting parameter, further model study should help determine the underlying physics of bacterial motility on surfaces.

## Discussion

Adhesion is crucial for bacterial infection and pathogenicity, followed by invasion or toxin injection. For several motile bacteria, external flagella could have a role of adhesin, such as H2 and H6 of enteropathogenic *E. coli* (EPEC) and H7 of enterohemorrhagic *E. coli* (EHEC)[41,42]. Flagella-mediated adhesion of the genus *Salmonella* is serovar-dependent, which is essential for serovars Enteritidis and Dublin but not for serovar Typhimurium[43]. In this study, we showed that flagella-independent adhesion of *Leptospira* bacteria allows their movement on kidney cells. Our experiments using clinical isolates and OMP-deficient mutants showed an inverse correlation between bacterial adhesion and crawling. The responsibility of motility for virulence was reaffirmed by plotting the asymptomatic and severely symptomatic pairs data on the inverse correlation between adherence and crawling. In addition, a simple numerical analysis suggested the significance of bacterial adhesion for mobility in the bacteria–host interface. There is an enormous combination between pathogenic leptospires and host species, suggesting that the pathogenic mechanism includes a complicated battle between bacteria and hosts, which involves immune response and gene regulation. An abundance of proteins in the outer membrane of pathogenic leptospires have different adhesion affinities to host cell components[32], and some of them, such as LigA examined in

speeds ($v_{CR}$) by the free-swimming speed ($v_{SW}$). $v_{CR}$ and $v_{SW}$ were averaged values of the speeds measured for individual bacteria (Supplementary Fig. S1). The error bars are the propagated errors yielded by calculations using average speeds and their standard errors (Supplementary Table S2 and Supplementary Note 1). The number of samples measured is shown in Supplementary Table S2.

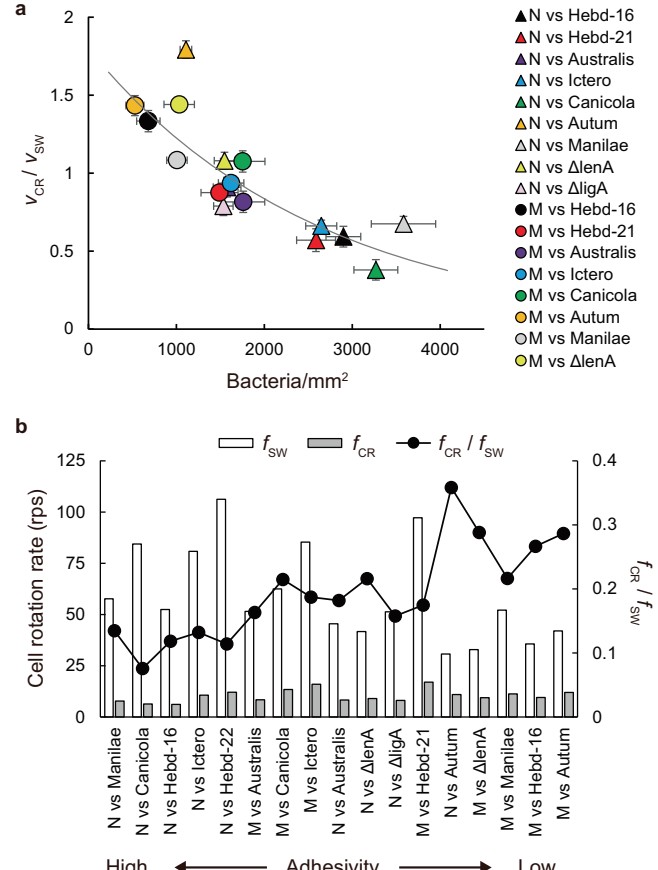

**Fig. 6 | Association of crawling and cell rotation with adhesion. a** Inverse correlation between crawling and adhesion. N and M represent NRK and MDCK, respectively. The solid line is the result of curve fitting. The correlation coefficient was −0.82. The error bars are the propagated errors yielded by calculations using average speeds and their standard errors (Supplementary Table S2 and Supplementary Note 1). Full names and the number of samples measured are shown in Supplementary Table S2. **b** The rotation rate of the leptospiral cell body during swimming ($f_{SW}$) and crawling ($f_{CR}$), calculated from the velocity and $(v/\lambda)/f$ (0.2 and 1 for swimming and crawling, respectively[13]). The values of $f_{CR}/f_{SW}$ are normalization of the burdened rotation speeds on the surface by those of free swimming.

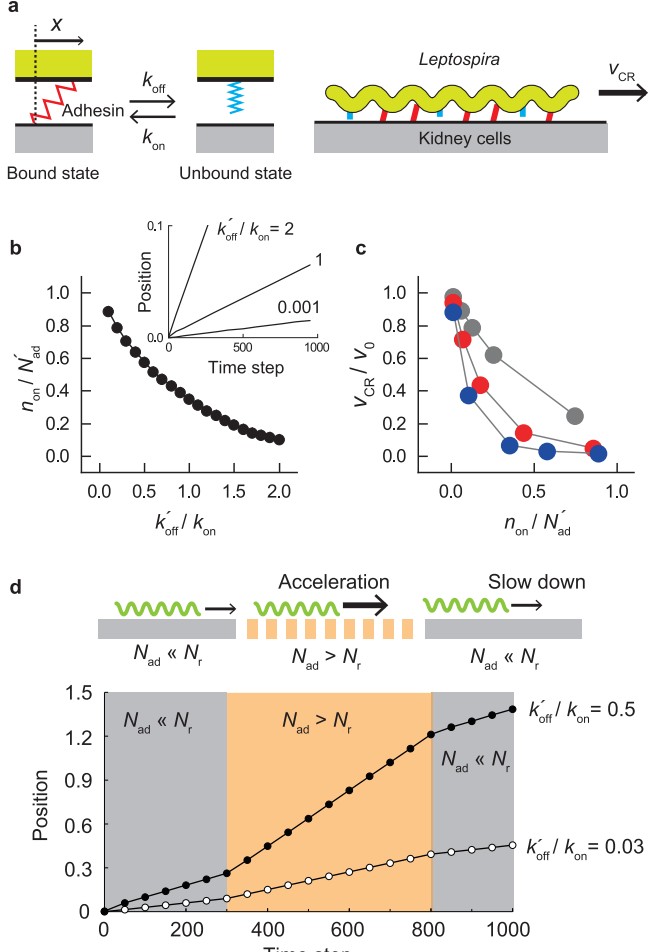

**Fig. 7 | Crawling model. a** Two-state model for crawling. Adhesins (e.g., OMPs and LPS) bind to kidney cells via a spring. The release of the adhesin-surface bond depends on the spring extension ($x$) by the bacterial movement. **b** Qualitative prediction of the dependence of the fraction of bound adhesins on $k'_{off}/k_{on}$. The inset shows accumulative displacements calculated by setting different values of $k'_{off}/k_{on}$. The averages of 10 runs are shown. **c** The dependence of $v_{CR}/v_0$ on $k'_{off}/k_{on}$. Based on the assumption of $N_{ad} \ll N_r$ (i.e., $N'_{ad} = N_{ad}$), calculations were performed using $N'_{ad} = 1000$ (gray), 2500 (red), or 5000 (blue). The averages of 10 runs are shown. **d** The upper schematic shows that a bacterium experiences a spatial change based on the density of host receptors ($N_r$). The lower panel demonstrates the change in the crawling speed with the variation of $N_r$. The simulations were performed using $k'_{off}/k_{on} = 0.5$ and 0.03 along with the temporal variation of $N_r$.

this study, show host-specific properties (Supplementary Fig. S2). Furthermore, although the present study focused on phenomena occurring a few hours after arrival at the organ, longer monitoring is needed for tangible discussions of pathogenesis and persistent infection in specific tissues. Identification of receptors on host cells involved in the adhesion and crawling of *Leptospira* spp. and their practical relevance to pathogenesis are future subjects.

None of our label-free experiments would have been successful without employing machine learning for motion analysis. Background subtraction has been a common process to detect intrusive objects in scene analysis of a surveillance system. We succeeded at introducing the computer vision technique to the in vitro bacterial infection assay, which can be applicable to diverse living cells in which probe labeling or genetic manipulations are difficult. In addition, the method will be useful for single molecule tracking to study membrane protein diffusion or turnover, which is generally difficult to detect due to low signal intensity, even if the target molecule is fluorescently labeled. There have been other methods developed to remove background noise from video data. In regard to their applications to infectious disease research, a method was developed to remove the background from video data of bacteria-derived scattering in patient biopsy by subtracting the minimum brightness on the time axis, the median

brightness of all frames, and the local spatial background[44]. This method does not require cumbersome sample pretreatment and yet enables rapid diagnosis and antibiotic susceptibility testing over a wide range of bacterial concentrations. Thus, the background subtraction method is expected to contribute to diverse research aimed at elucidating the living system.

## Methods
### Bacteria and animal cells
Bacterial strains used in this study are listed in Supplementary Table S1. The *Leptospira* strains except *L. interrogans* serogroup Autumnalis strain D-SA11-5K were cultured in Ellinghausen–McCullough–Johnson–Harris (EMJH) liquid medium[45] at 30 °C for 3–5 days; the Autumnalis strain was cultured in modified Korthof's liquid medium[46] at 30 °C for 3–5 days. *E. coli* β2163 was grown in LB medium supplemented with 0.3 mM diaminopimelate[47]. The

mammalian kidney epithelial cell lines used were MDCK-NBL2 (dog) and NRK-52E (rat). Kidney cells were maintained in Eagle's minimum essential medium (MEM) (Sigma-Aldrich, Darmstadt, Germany) for MDCK and Dulbecco's modified Eagle's medium (DMEM) (Thermo Fisher Scientific, MA, United States) for NRK containing 10% fetal bovine serum (Nacalai Tesque, Kyoto, Japan) and 1% antibiotic/anti-mycotic mixed solution (Nacalai Tesque) at 37 °C and 5% $CO_2$.

## Transposon insertion mutants

*lenA::Tn* and *ligA::Tn* mutants were obtained by random insertion mutagenesis of the *L. interrogans* strain UP-MMC-NIID using *Himar1*, which was conducted by conjugation with *E. coli* β2163 harboring pCjTKS2, as described previously[48]. After conjugation, cells were spread on a plate of EMJH agar (1.4% Bacto Agar, BD, USA) containing kanamycin at a final concentration of 25 µg/mL using glass beads (Bac'n'Roll Beads, Nippon Gene, Japan). The transposon insertion site was identified using semi-random PCR[48] with some modifications: the concentration of primers for the first and second PCR was reduced to 0.2 µM. The transposon was inserted into 259 bp of the 738-bp *lenA* gene and 3154 bp of the 3675-bp *ligA* gene, respectively.

## Expression and purification of recombinant GST/LenA fusion protein and production of anti-LenA serum

The DNA fragment corresponding to the amino acid position 27-241 of LenA from *L. interrogans* UP-MMC-NIID (WP_000809541) was amplified with PrimeSTAR GXL DNA polymerase (TaKaRa Bio) by using the upstream primer GSTSal-LenA-F (5'-CCCGGAATTCCCGGGTCAA-GAAGCGCAGATCTTAGG-3') and the downstream primer GSTSal-LenA-R (5'-ATGCGGCCGCTCGAGTCCGAATTTTCCCGCAAGTG-3'). After an initial 1 min denaturation at 95 °C, the reaction mixture was subjected to 30 cycles of denaturation at 98 °C for 10 s, annealing at 55 °C for 15 s, and extension at 72 °C for 1 min. The PCR product was purified, and cloned into *Sal*I-digested GST vector pGEX-6P-1 (Cytiva) by NEBuilder HiFi DNA Assembly cloning (New England BioLabs). To express the GST fusion protein, the plasmid was transformed into *E. coli* BL21. GST fusion protein expressed in *E. coli* BL21 was purified using glutathione Sepharose 4B (Cytiva) according to the supplier's instruction. For the production of antiserum against LenA, 30 µg of purified GST/LenA fusion protein was subjected to SDS-PAGE, from which the corresponding band was excised. The excised gel was injected into a female mouse with complete Freund's adjuvant and giving booster injections of the same amount of protein without adjuvant 2, 4, and 6 weeks later. The mouse was maintained at 23 ± 2 °C with 55 ± 5% humidity in a 12 h-light/12 h-dark cycle. The production of antiserum for LenA was approved by the animal research committee of National Institute of Infectious Diseases.

## SDS-PAGE and immunoblotting experiments

About $1.5 \times 10^8$ cells of wild-type and *lenA::Tn* and *ligA::Tn* mutant cells suspended in SDS-PAGE sample buffer were subjected to SDS-PAGE and Western blotting. The blot was incubated with antisera raised against LenA described above and the C-terminal portion of LigA[49]. The dilutions of antisera were 500-fold and 1000-fold for anti-LenA and anti-LigA, respectively, and the secondary antibody, goat anti-mouse IgG (H+L)-HRP conjugate (Bio-Rad Labotatories, Inc.), was diluted at 1:25,000 (Supplementary Fig. S3).

## Motility and adhesion assays

The cell density of the cultured leptospires were adjusted to $1 \times 10^8$ bacteria/mL (OD$_{420}$ ~0.1) using cultured media (EMJH or Korthof's medium). The bacterial suspension (20 µL) was infused to a chamber slide (SCS-N04, Matsunami Glass Ind., Ltd., Japan) where kidney cell monolayers were constructed as described previously[16]. After incubation for 15 min at 23 °C, *Leptospira* cells were observed using a dark-field microscope (BX53, ×40 UPlanFL N, ×5 relay lens, Olympus,

Tokyo, Japan) and recorded using a CCD camera (WAT- 910HX, Watec Co., Yamagata, Japan) at a frame rate of 30 Hz. The number of bacteria attached to the kidney cells were counted, and then swimming cells in liquid phase and crawling cells on kidney cells were measured using the image analytical program written in-house with Python.

## Background subtraction

The background model estimation method was based on ref. 20 with some modifications. Assuming a statistical model representing the background image (BG) without intrusive objects, we determined the appearance of masses of pixels that deviated from the modeled BG as intrusive foreground objects (FG).

Consider a single pixel value in grayscale $x^{(t)}$ at time $t$ and a training set $H_T = \{x^{(t)}, \cdots, x^{(t-T)}\}$ at time interval $T$. Given that the data set could possibly contain values corresponding to not only BG but also FG, the estimated probability density function from $H_T$ is denoted as $\hat{p}(x^{(t)}|H_T, BG+FG)$. Such pixel value distribution comprising multiple components can be modeled using multi-peak Gaussian (GMM):

$$\hat{p}(x|H_T, BG+FG) = \sum_{m=1}^{M} \hat{\pi}_m \mathcal{N}(x; \hat{\mu}_m, \hat{\sigma}_m^2), \tag{1}$$

where $m$ is the number of each component in the order of the weighting parameter $\hat{\pi}_m$ ($\sum_{m=1}^{M} \hat{\pi}_m = 1$), $M$ is the total number of components (i.e., $m = 1 - M$), and $\hat{\mu}_m$ and $\hat{\sigma}_m^2$ are the estimated values of the mean and variance, respectively. A new pixel value obtained at time $t+1$, $x^{(t+1)}$, was classified based on the Mahalanobis distance, and GMM parameters are updated using the following equations (Fig. 2c, d):

$$\hat{\pi}_m \leftarrow \hat{\pi}_m + \alpha(o_m^{(t)} - \hat{\pi}_m) - \alpha c_T, \tag{2}$$

$$\hat{\mu}_m \leftarrow \hat{\mu}_m + o_m^{(t)}(\alpha/\hat{\pi}_m)\delta_m, \tag{3}$$

$$\hat{\sigma}_m^2 \leftarrow \hat{\sigma}_m^2 + o_m^{(t)}(\alpha/\hat{\pi}_m)(\delta_m^2 - \hat{\sigma}_m^2). \tag{4}$$

In short, the distribution at time $t+1$ is predicted based on a set of images obtained before $t$. Repeating data acquisition and parameter estimation corresponds to learning a model for the scene. The decay parameter $\alpha(=0.001)$ affects the detection performance of bacteria. The background subtraction method recognizes a moving foreground against a static background. $\alpha$ is one of the parameters that determine the 'time limit' for being recognized as the foreground: If a bacterium remains immobile longer than the time limit, it is recognized as background, which is the limitation of this analysis (Supplementary Note 2 and and Supplementary Fig. S4). $\delta_m$ is the Euclidian distance from the mean of the cluster $m$ ($= x^{(t)} - \hat{\mu}_m$). $o_m^{(t)}$ determines the cluster where the new data is classified by setting $o_m^{(t)} = 1$ for the closest cluster and $o_m^{(t)} = 0$ for others. When the new data does not belong to any existing cluster, a new cluster $m = M + 1$ is generated with $\hat{\pi}_{M+1} = \alpha$, $\hat{\mu}_{M+1} = x^{(t)}$, and $\hat{\sigma}_{M+1} = \sigma_0$. Equation (2) was obtained using MAP estimation[20]. MAP $P(\boldsymbol{\pi}|\mathbf{n})$ is determined by a likelihood function $P(\mathbf{n}|\boldsymbol{\pi})$ and a prior distribution $P(\boldsymbol{\pi})$ as follows:

$$\arg \max_{\boldsymbol{\pi}} P(\boldsymbol{\pi}|\mathbf{n}) = \arg \max_{\boldsymbol{\pi}} P(\mathbf{n}|\boldsymbol{\pi})P(\boldsymbol{\pi}), \tag{5}$$

where $\mathbf{n} = \{\mathbf{n}_1, \mathbf{n}_2, \cdots, \mathbf{n}_T\}$ ($\mathbf{n}_t = \{o_1^{(t)}, o_2^{(t)}, \cdots, o_M^{(t)}\}$) indicates to which of the $m$ components the $t$-th data belongs. For example, the Mahalanobis distance determined the classification of the $t$-th data to $m = 1$, $o_1^{(t)} = 1$ and $o_{2 \sim M}^{(t)} = 0$ as described above. $P(\mathbf{n}|\boldsymbol{\pi})$ was a multinominal distribution:

$$P(\mathbf{n}|\boldsymbol{\pi}) = \prod_{i=1}^{T} \frac{1!}{\prod_{m=1}^{M} o_m^{(i)}!} \prod_{m=1}^{M} \hat{\pi}_m^{o_m^{(i)}}. \tag{6}$$

Therefore, the Dirichlet distribution was set as $P(\boldsymbol{\pi})$:

$$P(\boldsymbol{\pi}) = \frac{1}{Z} \prod_{m=1}^{M} \hat{\pi}_m^{c_m}, \qquad (7)$$

where $Z$ is the beta function. The hyperparameter $c_m$ was set in the range of $-1 < c_m < 0$, assuming GMM comprising the dominant component with large $\hat{\pi}_m$ (i.e., BG signal from cultured cells) and the other minor components with small $\hat{\pi}_m$ (i.e., FG signal mainly from bacteria) (Supplementary Note 3 and Supplementary Fig. S5). In Eq. (2), $c_T = -c_m/T$.

### Extraction of the targeted segments
Following the background subtraction processed by the pixels, the masses of pixels (segments) expected to be bacteria were extracted from the binarized images. First, dotted holes were filled by the dilation process, and then noise objects were removed using the erosion process. Next, the segments were screened according to area and aspect ratio based on the bacterial size (5–20 μm in length and ~150 nm in width). The aspect ratio was determined by ellipse fitting, and segments with <2 were excluded from the tracking objects. Migration velocities determined by the background subtraction method developed in this study were consistent with the results obtained by a conventional method, and the robustness of the present method against image quality (contrast and noise) was verified (Supplementary Note 4 and Supplementary Fig. S6).

When bacterial density is high, the overlapping of bacteria increases, and images at the moment of overlap are excluded by Step 4 of Fig. 2. Since the present study targets moving bacteria, the bacterial overlapping is temporary, and thus the tracking data before and after the overlap can be used for the measurement of motility. However, if the density of bacteria is extremely high and they are moving very slowly (the overlapping time is longer than the time limit discussed in Supplementary Fig. S4), the overlapping area could be recognized as background and excluded from the target of measurement. This problem did not occur under the present experimental condition (~$1 \times 10^8$ bacteria/mL), but care should be taken in experiments using high bacterial densities.

### Tracking of bacteria
Data set of object centroids obtained by ellipse fitting were compared to associate the same objects between frames. Consider consecutive frames $k$ containing $m$ objects and $k+1$ containing $n$ objects, having a data set of centroids $a_{k,n} = (x_{k,n}, y_{k,n})$ and $a_{k+1,m} = (x_{k+1,m}, y_{k+1,m})$, respectively. Calculating the square of distance $|a_{k,n} - a_{k+1,m}|^2$ for $n \times m$ pairs, identifying the closest pair as the same bacteria. Swimming and crawling speeds and MSD were obtained from locomotion trajectories of individual bacteria. Speeds were determined by line fitting to accumulative displacements of the bacterial centroids.

### Simulation
$\sum_{i=1}^{N} x_i$ at the $(j+1)$th event was stochastically given by determining the bound or unbound state for individual adhesins ($i = 1 \sim N$) at the $j$th. Random numbers were generated for each adhesin, and whether the bound adhesin at $j$ gets to the unbound state at $j+1$ and vice versa was determined based on $k_{on}$ and $k_{off}(x)$. $k_{off}$ of $x_{i,j}$ depends on $v_{CR}$ at $j$.

### Reporting summary
Further information on research design is available in the Nature Portfolio Reporting Summary linked to this article.

### Data availability
The raw tracking data of individual bacterial locomotion have been deposited in figshare with the identifier https://doi.org/10.6084/m9.figshare.24464290. Source data are provided with this paper.

## Code availability
The computer codes used for this study have been deposited in GitHub (https://github.com/shuichi-nakamura-e8/Label-free-bacterial-motion-tracking.git).

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

## Acknowledgements

We thank Dr. J. Xu (University of the Ryukyus) for his technical support and Dr. S. Toyabe (Tohoku University) for the insightful discussion. This work was supported by the JSPS KAKENHI: 21H02727 for S.N. and 19K07571 and 22K07062 for N.K.

## Author contributions

K.A., N.K., and S.N. planned the project and wrote the manuscript. K.A., N.K., and S.N. carried out the experiments and data analysis. K.A. and S.N. set up the optical system. K.A. made programs for data analysis. All authors contributed to the article and approved the submitted version.

## Competing interests

The authors declare no competing interests.
