## [Peer Review File · Nature Communications]

Machine learning-based motion tracking reveals an inverse correlation between adhesivity and surface motility of the leptospirosis spirocheteReviewer #1 (Remarks to the Author):

The manuscript reports a label-free motion tracking method for study bacteria motility on cultured mammalian cells, and the application of the method on study leptospiral strains from different sources including clinical isolates. The method uses a Gaussian mixture model (GMM) to identify moving bacterial cells from the complexed scattering background of the cultured mammalian cells in darkfield optical image sequences. While both darkfield optical imaging of bacteria motion and using GMM for background subtraction is not new, the application of these methods to analysis bacteria motion in cell culture is innovative and practical. Compares to fluorescence-based tracking method, this label-free method allows tracking of any bacteria species without genetically encoding fluorescence proteins, which is not always feasible as the authors experiences. The manuscript is well presented with sufficient details, and the conclusion is supported by the data. I only have the following question:

Does the concentration/density of bacteria affect the machine learning based tracking method? Since the GMM based background subtraction method is assuming the static background of the culture mammalian cells are dominant the images, will a high density of moving bacterial cells has an impact on the background removal and bacteria tracking accuracy?

Reviewer #2 (Remarks to the Author):

In this study, the authors have developed label-free tracking tools to monitor the movement of *Leptospira* sp. on the surface of kidney cells. Their objective is to investigate potential correlations between motility and adhesion behaviors and the pathogenicity of these bacteria.

While there are some intriguing aspects to this study, I believe they require further expansion to justify publication in a generalist journal:

Tracking Methodology and Its Applicability: Tracking bacterial movements in complex environments with a low signal-to-noise ratio is a challenging endeavor. The study explores AI-based methods as a solution, which is commendable. However, it's essential to discuss the broader applicability of this methodology. Can it be adapted for use in other systems, and for what specific purposes? Additionally, the study should address the limitations of this method, including the types of errors it may introduce and how these errors affect the study's conclusions. While the authors provide a theoretical limit to tracking, it would be valuable to compare this to ground truth experiments and present a statistical analysis of the error. Numerous established methods exist for such validations.

Biological Conclusions: The biological conclusions drawn in this study appear somewhat limited and mostly descriptive. They lack significant biological insights and seem more like an initial description of *Leptospira* movements on kidney cell surfaces. This work could serve as groundwork for future research, but the current biological findings are either speculative or self-evident:

a. **Correlation of Movement Speed and Cell Surface Residence:** The claim that movement speed and cell surface residence correlate with symptomatic/asymptomatic phenotypes appears bold and oversimplified. Symptoms in this context likely result from complex interactions at the physiopathological level. It is doubtful that these outcomes can be solely explained by the simple behaviors observed in the authors' experimental setup.

b. **Adhesion and Motility:** The observation that adhesion is inversely related to motility is not novel and has been observed in various systems, especially in biofilm development. It is not very clear also what insight the crawling model provides. While studying this phenomenon in *Leptospira* on kidney cells is important, it's crucial to recognize that this is an initial observation requiring further mechanistic insights. The examination of OMP mutants is a start, but it remains unclear how

specific the targeted genes are and how broadly envelope defects manifest when disrupted. Consequently, the conclusions drawn from this analysis do not provide substantially new information beyond recognizing that envelope mutants exhibit impaired crawling.

In conclusion, this study possesses a solid foundation and presents an intriguing methodology, but the drawn conclusions either lack novelty or sufficient justification. Consequently, it is unclear how this work could impact a non-specialist community. Further research, expanded discussion, and a more comprehensive analysis of the biological implications are needed to strengthen the relevance of this study in a generalist journal.

Reviewer #3 (Remarks to the Author):

This is a very important and novel contribution to the leptospiral motility literature. The authors describe two very different motility patterns that vary depending on the type of host cell (NRK vs MDCK) and leptospiral strain. The following comments are intended to improve the manuscript:

It would be helpful to describe in layman's terms how "machine-learning" was used to track leptospiral motion. I think of "machine learning" as having two phases: training phase and a test phase. In this study, what data were used for training the tracking algorithm? Please clarify.

Although "confined crawling" vs "swimming" seem like apt descriptions, I disagree with the description of "confined crawling on reservoir host cells". It seems entirely possible, if not likely, that at least some of the confined crawling movements could occur either underneath or inside of host cells, particularly since the pathways appear to be "confined" (I attached an image showing a leptospire with a confined (ie constrained) mobility path during seconds 19-23 of the video. I doubt that the approach used here can distinguish the actual location (on top of, underneath, or inside) of the leptospire relative to the host cells. A recent study by Santecchia et al. (Front Cell Infect Microbiol 2022) found that leptospire readily enter and exit host cells very dynamically. Perhaps "confined crawling in association with reservoir host cells" would be more accurate.

My interpretation of the reason that the asymptomatic pair exhibited slower crawling velocities on NRK than MDCK cells is that *L. interrogans* sv *Manilae* is more adapted to rat cells in terms of stronger adhesion binding to host cell ligands, which would slow down movements. This is consistent with the lower adhesion to NRK cells and increased crawling velocities in association with NRK cells of *lenA* and *ligA* mutants.

In the histograms in Fig. 3b, what does "PDF" refer to in the y-axis? Please explain this in the figure legend.

Reviewer #3 Attachment on the following page

00:21 | 00:23

Navigation controls: play/pause, stop, previous, next, and a progress bar.

Response to Reviewers' comments

To Reviewer #1:

The manuscript reports a label-free motion tracking method for study bacteria motility on cultured mammalian cells, and the application of the method on study leptospiral strains from different sources including clinical isolates. The method uses a Gaussian mixture model (GMM) to identify moving bacterial cells from the complexed scattering background of the cultured mammalian cells in darkfield optical image sequences. While both darkfield optical imaging of bacteria motion and using GMM for background subtraction is not new, the application of these methods to analysis bacteria motion in cell culture is innovative and practical. Compares to fluorescence-based tracking method, this label-free method allows tracking of any bacteria species without genetically encoding fluorescence proteins, which is not always feasible as the authors experiences. The manuscript is well presented with sufficient details, and the conclusion is supported by the data. I only have the following question:

Does the concentration/density of bacteria affect the machine learning based tracking method? Since the GMM based background subtraction method is assuming the static background of the culture mammalian cells are dominant the images, will a high density of moving bacterial cells has an impact on the background removal and bacteria tracking accuracy?

Re) We appreciate your recognition of the value of this study. Your point is very important in terms of the accuracy of the analysis. As you understand, when bacterial density is high, the number of cases of overlapping bacteria increases. Since this study targets moving bacteria, the overlapping of bacteria is temporary and does not cause a problem. Images at the moment of overlap are excluded in steps 3-4 of Fig. 2, but the tracking data before and after the overlap can be used. However, if the density of bacteria is extremely high and they are moving very slowly (the overlap time is longer than the "time limit" determined by the parameters discussed on page 6 of Supplementary Information), the overlapping area may be considered as background and excluded from the measurement. Such a problem did not occur under the conditions of the present experiments (density of bacteria to be infected), but care should be taken in experiments in which bacteria are infected at very high densities.

We added a description of the potential effects of high bacterial concentrations on the method's accuracy to the Methods section:

(p7, right column, line 39) *When bacterial density is high, the overlapping of bacteria increases, and images at the moment of overlap are excluded by Step 4 of Figure 2. Since the present study targets moving bacteria, the bacterial overlapping is temporary, and thus the tracking data before and after the overlap can be used for the measurement of motility. However, if the density of bacteria is extremely high and they are moving very slowly (the overlapping time is longer than the time limit discussed in Figure S4), the overlapping area could be recognized as background and excluded from the target of measurement. This problem did not occur under the present experimental condition (1×10^8 bacteria/mL), but care should be taken in experiments in which bacteria are infected at higher densities.*

To Reviewer #2:

In this study, the authors have developed label-free tracking tools to monitor the movement of *Leptospira* sp. on the surface of kidney cells. Their objective is to investigate potential correlations between motility and adhesion behaviors and the pathogenicity of these bacteria.

While there are some intriguing aspects to this study, I believe they require further expansion to justify publication in a generalist journal:

Tracking Methodology and Its Applicability: Tracking bacterial movements in complex environments with a low signal-to-noise ratio is a challenging endeavor. The study explores AI-based methods as a solution, which is commendable. However, it's essential to discuss the broader applicability of this methodology. Can it be adapted for use in other systems, and for what specific purposes? Additionally, the study should address the limitations of this method, including the types of errors it may introduce and how these errors affect the study's conclusions. While the authors provide a theoretical limit to tracking, it would be valuable to compare this to ground truth experiments and present a statistical analysis of the error. Numerous established methods exist for such validations.

Re) We appreciate your very important remarks.

First, we will answer the application of this image analysis technology. The background subtraction method has been used in various situations such as detection of moving objects in surveillance cameras and detection of contamination of objects in factories. The novelty of this study is that we extended this technique to life science experimental techniques and succeeded in detecting specific objects in noisy images with a background of cell images in liquids. This method is expected to be further utilized in the life science field. For example, our background subtraction method is useful for single molecule tracking to study membrane protein diffusion and turnover, which is generally difficult to detect due to low signal intensity, even if the target molecule is fluorescently labeled. Since such specific examples are significant to show the extensibility of our method, we have added the following text in the last paragraph of the main text :

(p6, right column, line 13) *We succeeded at introducing the computer vision technique to the in vitro bacterial infection assay, which can be applicable to diverse living cells, in which probe labeling or genetic manipulations are difficult. In addition, the method will be useful for single molecule tracking to study membrane protein diffusion or turnover, which is generally difficult to detect due to low signal intensity, even if the target molecule is fluorescently labeled. Thus, the background subtraction method is expected to contribute to a wide range of research aimed at elucidating living system.*

Next, we respond regarding errors. As the reviewer mentioned, it is very critical to understand the possibility of errors. The reviewer suggested comparing the theoretical limit with the ground-truth experiments, which is the primary method of standard supervised learning. However, as explained below, this error verification cannot in principle be performed for the background subtraction method in this study.

Simply stated, when the distribution will be predicted at time t , then a set of images taken prior to t is learned in the background subtraction method. By repeating the estimation of the Gaussian parameters that match the acquired data sequentially, the optimal model for the scene is predicted sequentially. That is, there is no such thing as correct data (ground-truth data) because the background subtraction method learns from results predicted before a certain point in time to predict the next result. While verification of errors is undoubtedly important, verification by comparison with correct data cannot be performed in the background subtraction method. In the case of surveillance cameras, where the background subtraction method is already in practical use, representative sample data, such as images of cars on the road, may be used in place of the correct data. However, since this study proposes a new application of the background subtraction method, there is no alternative to such generic sample data. The only possible alternative is to artificially create correct and incorrect data by us. Then, we analyze the artificial data with our program, draw ROC curves, and evaluate them with AUC. However, this method is not objective because of the arbitrariness involved in creating the artificial data, and the evaluation results have no value. For these reasons, the background subtraction method in this study cannot, in principle, perform error evaluation used in general machine learning. However, as the reviewer points out, it is very important to make explicit the limitations of our method. Therefore, we would greatly appreciate it if you could acknowledge this matter by modifying the METHODS to better emphasize the limitations of this method:

(p6, right column, line 2) *The decay parameter α (= 0.001) affects the detection performance of bacteria. The background subtraction method recognizes a moving foreground against a static background. α is one of the parameters that determine the "time limit" for being recognized as the foreground: If a bacterium remains immobile longer than the time limit, it is recognized as background, which is the limitation of this analysis (Figure S4).*

Biological Conclusions: The biological conclusions drawn in this study appear somewhat limited and mostly descriptive. They lack significant biological insights and seem more like an initial description of *Leptospira* movements on kidney cell surfaces. This work could serve as groundwork for future research, but the current biological findings are either speculative or self-evident:

a. Correlation of Movement Speed and Cell Surface Residence: The claim that movement speed and cell surface residence correlate with symptomatic/asymptomatic phenotypes appears bold and oversimplified. Symptoms in this context likely result from complex interactions at the physiopathological level. It is doubtful that these outcomes can be solely explained by the simple behaviors observed in the authors' experimental setup.

Re) We agree with the reviewer on this point and have modified the description of the Conclusion, based on the reviewer's comments. We have weakened our argument by specifically noting the phenomena not addressed in this study. We hope that the following modification also serves as a response to the next comment:

(p6, left column, line 55) *There is an enormous combination between pathogenic leptospires and host species, suggesting that the pathogenic mechanism includes a complicated battle between bacteria and hosts, which involves immune response and gene regulation. An abundance of proteins in the outer membrane of pathogenic leptospires have different adhesion affinities to host cell components (Robbins et al., 2015, ref. 32), and some of them, such as LigA examined in this study, show host-specific properties (Figure S2). Furthermore, although the present study focused on phenomena occurring a few hours after arrival at the organ, longer monitoring is needed for tangible discussions of pathogenesis and persistent infection in specific tissues. Identification of receptors on host cells involved in the adhesion and crawling of Leptospira spp. and their practical relevance to pathogenesis are future subjects.*

b. Adhesion and Motility: The observation that adhesion is inversely related to motility is not novel and has been observed in various systems, especially in biofilm development. It is not very clear also what insight the crawling model provides. While studying this phenomenon in *Leptospira* on kidney cells is important, it's crucial to recognize that this is an initial observation requiring further mechanistic insights. The examination of OMP mutants is a start, but it remains unclear how specific the targeted genes are and how broadly envelope defects manifest when disrupted. Consequently, the conclusions drawn from this analysis do not provide substantially new information beyond recognizing that envelope mutants exhibit impaired crawling.

In conclusion, this study possesses a solid foundation and presents an intriguing methodology, but the drawn conclusions either lack novelty or sufficient justification. Consequently, it is unclear how this work could impact a non-specialist community. Further research, expanded discussion, and a more comprehensive analysis of the biological implications are needed to strengthen the relevance of this study in a generalist journal.

Re) As the reviewer points out, the results of this study cannot explain all disease mechanisms at the molecular level. However, the fact that the relationship between adhesion and motility, which has been observed in various scenes such as biofilms, was also observed in the establishment of infection is significant as a common principle in biology. We believe that our results, the inverse correlation between adhesion and motility, give insight into the first step of persistent infection (establishment of carrier status). It is also important to note that among the many outer membrane proteins, we found one that had a particularly strong effect on adhesion to dog cells, a result that could not have been obtained without the background subtraction method and mutant analysis in this study. However, it remains unknown about physical changes in outer membrane of *L. interrogans* caused by the loss of these proteins, and their receptors on the host cells have not yet been identified. To clarify this point, we have revised the conclusion as indicated in response to the above comment. It is true that without elucidation of these issues, it will be impossible to understand the full extent of the disease or to develop practical applications, such as vaccines. Our group is currently working on these questions, and we hope that you will look forward to our future research.

To Reviewer #3:

This is a very important and novel contribution to the leptospiral motility literature. The authors describe two very different motility patterns that vary depending on the type of host cell (NRK vs MDCK) and leptospiral strain. The following comments are intended to improve the manuscript:

It would be helpful to describe in layman's terms how "machine-learning" was used to track leptospiral motion. I think of "machine learning" as having two phases: training phase and a test phase. In this study,

what data were used for training the tracking algorithm? Please clarify.

Re) We appreciate your acknowledgment of the significance and novelty of this study. We indeed understand that it is difficult to grasp "what is being learned" by this machine learning. The background-subtraction method estimates the Gaussian distribution, i.e., background or bacteria, sequentially for each pixel. Simply stated, if the distribution is predicted at time t , then a set of images taken prior to t is being learned. By repeating the estimation of the Gaussian parameters that match the acquired data sequentially, we predict the optimal model corresponding to the scene sequentially. To explain this simply, the following sentences have been added to the Methods section:

(p7, left column, line 60) *In short, the distribution at time $t+1$ is predicted based on a set of images obtained before t . Repeating data acquisition and parameter estimation corresponds to learning a model for the scene.*

Although "confined crawling" vs "swimming" seem like apt descriptions, I disagree with the description of "confined crawling on reservoir host cells". It seems entirely possible, if not likely, that at least some of the confined crawling movements could occur either underneath or inside of host cells, particularly since the pathways appear to be "confined" (I attached an image showing a leptospire with a confined (ie constrained) mobility path during seconds 19-23 of the video. I doubt that the approach used here can distinguish the actual location (on top of, underneath, or inside) of the leptospire relative to the host cells. A recent study by Santecchia et al. (Front Cell Infect Microbiol 2022) found that leptospire readily enter and exit host cells very dynamically. Perhaps "confined crawling in association with reservoir host cells" would be more accurate.

Re) Thank you for pointing this out. Since *Leptospira* moves three-dimensionally through the kidney cell layer, the phrase "confined crawling in association with reservoir host cells" suggested by the reviewer is appropriate. We have corrected it (p3, left column, lines 8 and 21).

My interpretation of the reason that the asymptomatic pair exhibited slower crawling velocities on NRK than MDCK cells is that *L. interrogans* sv Manilae is more adapted to rat cells in terms of stronger adhesin binding to host cell ligands, which would slow down movements. This is consistent with the lower adhesion to NRK cells and increased crawling velocities in association with NRK cells of *lenA* and *ligA* mutants.

Re) Thank you for your new perspective. We did not have the viewpoint of adaptation to rat cells, but such an interpretation certainly seems reasonable. We have added the following statement to the section of "Crawling model":

(p6, left column, line 2) *The two-state model assuming $N'_{ad} = N_{ad}$ qualitatively demonstrated that an increment of k'_{off}/k_{on} decreased the fraction of the bound adhesins non/N'_{ad} (Figure 7b), affecting v_{CR}/v_0 anti-correlatively (Figure 7c). *L. interrogans* serovar Manilae might be adapted to rats in terms of stronger adhesin binding to the ligands on the kidney cells. We also found that the reduced number of OMPs by gene knockout facilitated crawling (Figure 5). The tendency was shown by the model in the N'_{ad} dependence of v_{CR}/v_0 (Figure 7c).*

In the histograms in Fig. 3b, what does "PDF" refer to in the y-axis? Please explain this in the figure legend.

Re) We apologize for the missing explanation. PDF is short for Probability Density Function. We added the explanation to the legend.

Reviewer #1 (Remarks to the Author):

The revision addressed most of the reviewers' questions for the original manuscript well.

Regarding the comments from reviewer 2 on comparing to ground truth data for statistical analysis of the error, I agree with the authors that the background removal method used in the paper is not a deep learning or supervised learning algorithm, but a statistic prediction model that not using any ground truth training dataset. However, as the authors agreed, it is still important to validate the method in some way. Is it possible to using the fluorescence labelled bacteria and fluorescence image as ground truth to validate the accuracy/error of the label-free background subtraction and tracking method presented here? As mentioned in the introduction, the authors have used GFP as label for some model strains (*Leptospira* spp.) so this seems is feasible at least for using this strain as a model for validation purpose.

Another related question I have is how computationally intensive the background removal method? Will simple subtractions of temporal local minimum be sufficient for removing the background? An example of the method is reported here: F. Zhang et.al., *Anal. Chem.* 2021, 93, 18, 7011–7021

Response to Reviewers' comments

To Reviewer #1:

Thank you very much for agreeing to review our revised manuscript. We also appreciate your very helpful advice on addressing the comment from Reviewer #2 in the first review. We will respond to your comments below.

- 1) Regarding the comments from reviewer 2 on comparing to ground truth data for statistical analysis of the error, I agree with the authors that the background removal method used in the paper is not a deep learning or supervised learning algorithm, but a statistic prediction model that not using any ground truth training dataset. However, as the authors agreed, it is still important to validate the method in some way. Is it possible to using the fluorescence labelled bacteria and fluorescence image as ground truth to validate the accuracy/error of the label-free background subtraction and tracking method presented here? As mentioned in the introduction, the authors have used GFP as label for some model strains (*Leptospira* spp.) so this seems is feasible at least for using this strain as a model for validation purpose.

Re) Thank you for your very helpful suggestions. In accordance with your comment, we have analyzed the GFP-labeled *Leptospira* using our background-subtraction method. We added the new results to Supplementary Information (Figure S6), which was referred to in the main text (p7, right column, lines 46-50). The verification showed that the migration speed determined by the background subtraction method was consistent with the results obtained by a conventional tracking method (i.e., cell centroid tracking) and that the present method is robust to the change in the image quality (contrast and noise). We hope that the new data will validate our methodology. We thank you once again for your helpful advice.

- 2) Another related question I have is how computationally intensive the background removal method? Will simple subtractions of temporal local minimum be sufficient for removing the background? An example of the method is reported here: F. Zhang et.al., Anal. Chem. 2021, 93, 18, 7011–7021

Re) Thank you for your question and for sharing interesting literature with us. Our background subtraction method can complete the analysis of a 30-second video in a dozen seconds. It is difficult to say whether the method in the paper you introduced can perform the same analysis as this study without implementing it into our program and verifying its operation in detail. However, the two methods have in common that they focus on the difference in brightness between the foreground and background and consider the area that has been captured for a longer time to be the background, so we assume that the background subtraction you mentioned may be useful for tracking label-free bacteria.

As you know, there are many algorithms for background subtraction. The variety of methods is very important in confronting unknown problems. In this revised manuscript we have referred to this by citing the paper you introduced (p6, right column, lines 21-31). We hope that this paper will be a catalyst for the development of more sophisticated analytical methods in the future.

Reviewer #1 (Remarks to the Author):

The second revision has addressed all my comments.